# *Salmonella* Vaccine Study in Oxford (SALVO) trial: protocol for an observer-participant blind randomised placebo-controlled trial of the iNTS-GMMA vaccine within a European cohort

Brama Hanumunthadu [ID],[1] Nasir Kanji,[1] Nelly Owino,[1] Carla Ferreira Da Silva,[1] Hannah Robinson,[1,2] Rachel White,[1] Pietro Ferruzzi,[3] Usman Nakakana,[3] Rocio Canals,[3] Andrew J Pollard [ID],[1] Maheshi Ramasamy,[1] Vacc-iNTS Consortium

[1]Department of Paediatrics, University of Oxford, Oxford Vaccine Group, Oxford, UK
[2]NIHR Oxford Biomedical Research Centre, Oxford, UK
[3]GSK Vaccines Institute for Global Health, Siena, Italy

**Correspondence to**
Dr Brama Hanumunthadu;
brama.hanumunthadu@paediatrics.ox.ac.uk

## ABSTRACT

**Introduction** Invasive non-typhoidal Salmonellosis (iNTS) is mainly caused by *Salmonella enterica* serovars Typhimurium and Enteritidis and is estimated to result in 77 500 deaths per year, disproportionately affecting children under 5 years of age in sub-Saharan Africa. Invasive non-typhoidal *Salmonellae* serovars are increasingly acquiring resistance to first-line antibiotics, thus an effective vaccine would be a valuable tool in reducing morbidity and mortality from infection. While NTS livestock vaccines are in wide use, no licensed vaccines exist for use in humans. Here, a first-in-human study of a novel vaccine (iNTS-GMMA) containing *S.* Typhimurium and *S.* Enteritidis Generalised Modules for Membrane Antigens (GMMA) outer membrane vesicles is presented.

**Method and analysis** The *Salmonella* Vaccine Study in Oxford is a randomised placebo-controlled participant-observer blind phase I study of the iNTS-GMMA vaccine. Healthy adult volunteers will be randomised to receive three intramuscular injections of the iNTS-GMMA vaccine, containing equal quantities of *S.* Typhimurium and *S.* Enteritidis GMMA particles adsorbed on Alhydrogel, or an Alhydrogel placebo at 0, 2 and 6 months. Participants will be sequentially enrolled into three groups: group 1, 1:1 randomisation to low dose iNTS-GMMA vaccine or placebo; group 2, 1:1 randomisation to full dose iNTS-GMMA vaccine or placebo; group 3, 2:1 randomisation to full dose or lower dose (dependant on DSMC reviews of groups 1 and 2) iNTS-GMMA vaccine or placebo. The primary objective is safety and tolerability of the vaccine. The secondary objective is immunogenicity as measured by O-antigen based ELISA. Further exploratory objectives will characterise the expanded human immune profile.

**Ethics and dissemination** Ethical approval for this study has been obtained from the South Central—Oxford A Research Ethics Committee (Ethics REF:22/SC/0059). Appropriate documentation and regulatory approvals have been acquired. Results will be disseminated via peer-reviewed articles and conferences.

**Trial registration number** EudraCT Number: 2020-000510-14.

## STRENGTH AND LIMITATIONS OF THIS STUDY

⇒ *Salmonella* Vaccine Study in Oxford is a first-in-human study of a novel vaccine against invasive non-typhoidal Salmonellosis (iNTS), a neglected disease largely affecting low- and middle-income countries.

⇒ This study provides the opportunity to investigate the adaptive immune response to bacterial outer membrane antigens, supporting future vaccine development efforts against this disease.

⇒ The volunteers recruited to this trial may not be immunologically representative of the ultimate target population in endemic areas. In addition, the vaccination schedule in this study is based on the expected schedule in infants.

⇒ In the absence of a known correlate of protection against iNTS disease, it will not be possible to determine vaccine efficacy in this trial.

## INTRODUCTION

Non-typhoidal *Salmonellae* (NTS), such as *Salmonella enterica* serovars Enteritidis and Typhimurium, most commonly cause a self-limiting gastroenteritis that is indistinguishable from that caused by many other enteric pathogens.[1] However, some NTS bacterial strains can also cause an invasive syndrome with bacteraemia, high fevers and metastatic infection which if untreated can lead to septicaemia and death. Invasive non-typhoidal *Salmonella* (iNTS) infections are more common in children, the elderly and in the immunosuppressed, including HIV-infected individuals.[2,3]

The Global Burden of Disease study estimates 535 000 annual cases of iNTS globally, associated with 77 500 deaths in 2017 alone, representing a higher case fatality rate when compared with non-typhoid *Salmonella*

gastroenteritis or typhoidal *Salmonella*.[1 4] The highest burden of iNTS disease globally occurs in sub-Saharan Africa, with a pooled annual incidence of 52/100 000.[1 4] This is likely an underestimate given the limited availability of diagnostics in the region coupled with non-specific disease presentation. The age at which infection occurs shows a bimodal distribution in most African studies with 68.3% occurring in children under 5 years and a second peak in the 30–40 years age group, believed to be associated with HIV, malaria and malnutrition.[3 5–7]

Of the invasive pathogens responsible for iNTS in sub-Saharan Africa, *S.* Typhimurium is implicated in approximately two-thirds of all cases, with the ST313 serotype accounting for most isolates.[8] In contrast to other non-invasive strains, African ST313 isolates often exhibit genomic degradation and pseudogene formation like that seen in typhoidal *Salmonellae*, which contribute to human host restriction and an invasive phenotype.[9–12] Furthermore, iNTS strains such as ST313 have been associated with multidrug resistance, leading to *Salmonellae* being classified as WHO high priority antibiotic resistant pathogens.[13–15]

High mortality, logistical difficulties in diagnosing infection in the developing world and increasing antimicrobial resistance strongly advocate for the development of an effective vaccine.

There are no currently licensed vaccines for iNTS although multiple candidates are in early phase development, including O-antigen (OAg) conjugates, oral attenuated vaccines and multiple antigen display protein-polysaccharide conjugate vaccines. A trivalent iNTS-typhoid vaccine is currently in phase I.[16]

The investigational product in this study is the iNTS-GMMA vaccine. This novel vaccine developed by GSK Vaccines for Global Health consists of outer membrane vesicles or Generalised Modules for Membrane Antigens (GMMA) of the two most common serotypes associated with invasive disease, *Salmonella* Enteritidis (SEn) and *Salmonella* Typhimurium (STm).[17] GMMA particles contain several immunodominant antigens including the OAg component of bacterial lipopolysaccharide and outer membrane proteins. iNTS-GMMA are immunogenic in animal models, eliciting antibodies directed against OAg and demonstrating serum bactericidal activity. Immunised animals also appear to have lower systemic bacterial loads on subsequent challenge.[18]

This is the first trial to investigate the iNTS-GMMA vaccine in humans. Demonstration of safety and immunogenicity in this study will lead to progression to subsequent studies in a sub-Saharan country of high endemicity.

### Study aims and objectives

The aim of the trial is to determine the safety of the iNTS-GMMA vaccine and study the immune response to vaccination. Primary, secondary and exploratory objectives are detailed in table 1.

As this is a first-in-human vaccine trial, the primary objective is the safety and tolerability of the iNTS-GMMA vaccine which will be ascertained by the collection of solicited and unsolicited adverse events, serious adverse events, withdrawals and laboratory parameters. Solicited adverse events will be collected up to 7 days following each vaccination and include local injection site reactions and systemic symptoms. Unsolicited adverse events will be collected up to 28 days following each vaccination, and outside of this period will be recorded only if medically attended. Serious adverse events and withdrawals will be recorded throughout the study. Laboratory adverse events will be recorded by obtaining blood samples to analyse full blood count, renal and liver profile at days 0, 7, 28, 56, 63, 84, 168, 175 and 196.

The secondary objective is the determination of immunogenicity of the iNTS-GMMA vaccine. Immunogenicity will be measured using SEn and STm O-antigen specific ELISA at time points days 0, 7, 28, 56, 63, 84, 168, 175, 196 and 350.

## METHODS
### Trial interventions: investigational medical product (IMP) and placebo

The iNTS-GMMA vaccine consists of outer membrane vesicles or GMMA from the two most common serovars causing invasive disease, *Salmonella* Enteritidis and S*almonella* Typhimurium adsorbed onto Alhydrogel (0.35 mg AL$^{3+}$/0.5 mL dose) and suspended in isotonic

**Table 1** Primary, secondary and exploratory objectives and outcomes of the SALVO study

| Objective | | Outcome measure |
|---|---|---|
| Primary | To determine the safety and tolerability of the iNTS-GMMA vaccine | Clinical review and participant recording of solicited, unsolicited adverse events, serious adverse events, withdrawals and laboratory parameters (haematology/biochemistry) |
| Secondary | To investigate immunogenicity of the iNTS-GMMA vaccine | Measurement of serovar specific SEn and STm O-antigen by ELISA before and after vaccination |
| Exploratory | To further characterise the immune response to vaccination | Exploratory immunological analyses including functional antibody assays and antigen-specific memory B cell responses and T cell responses before and after vaccination |

GMMA, Generalised Modules for Membrane Antigens; iNTS, invasive non-typhoidal *Salmonellosis*; SALVO, *Salmonella* Vaccine Study in Oxford; SEn, *Salmonella* Enteritidis; STm, *Salmonella* Typhimurium.

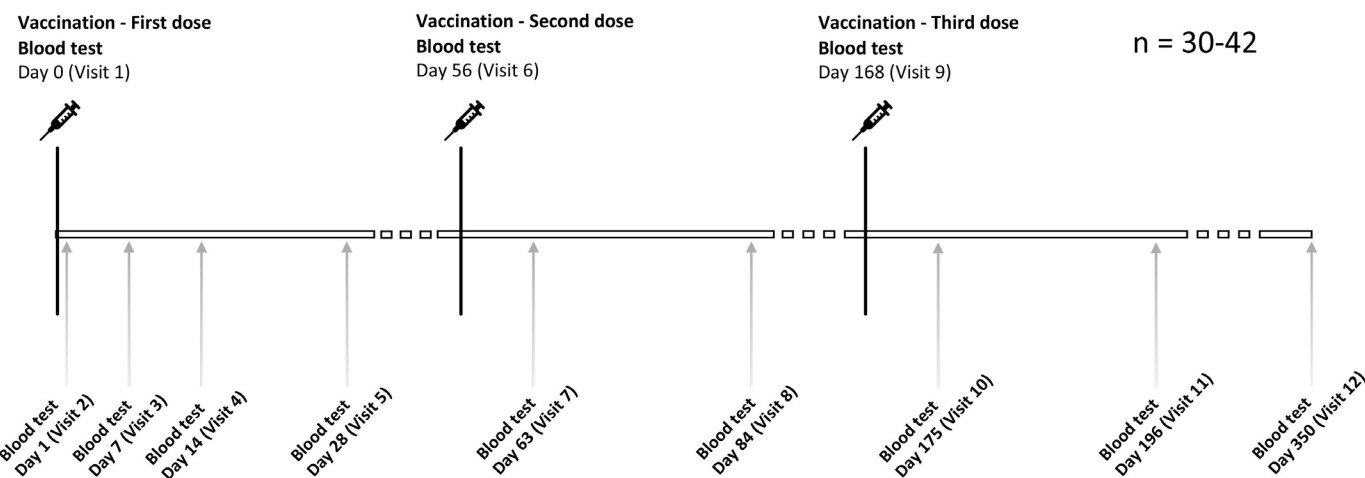

**Figure 1** *Salmonella* Vaccine Study in Oxford vaccine and visit schedule.

phosphate buffered saline. The parent bacteria have been genetically modified to increase production of outer membrane vesicles ($\Delta tolR$) and reduce the toxicity of lipid A component within the lipopolysaccharide ($\Delta msbB$ and $\Delta pagP$). The GMMA particles are filtered and purified to form the active component of the vaccine.[19–21] Two dose levels will be used for this study: a full dose of 20 µg STmGMMA+20 µg SEnGMMA (total 40 µg OAg); and a lower dose consisting of 5.3 µg STmGMMA+5.3 µg SEnGMMA (total 10.6 µg OAg). The placebo matches the vaccine matrix and consists of Alhydrogel without a GMMA component. The vaccine and placebo are both administered as intramuscular injections.

### Study design and setting

This is a first-in-human randomised placebo-controlled participant-observer blind trial of the iNTS-GMMA vaccine in healthy adults aged 18–55 years in the UK. A total of 30–42 participants will be randomised to receive three intramuscular doses of active vaccine or placebo at 0, 2 and 6 months (figure 1). For further details, please see SALVO Protocol in online supplemental material 1.

Participants will be sequentially enrolled into three groups (figure 2) with a dose escalation between group 1 (lower dose iNTS-GMMA vaccine, 10.6 µg total OAg content) and group 2 (full dose iNTS-GMMA vaccine, 40 µg total OAg content). These first two groups will each consist of six participants who will be randomised 1:1 to the active vaccine or placebo. An additional six participants may be recruited to each of these groups if further safety information is required. Group 3 consists of 18 participants randomised 2:1 to receive the iNTS-GMMA vaccine or placebo. The decision to proceed to low or full dose vaccine in group 3 will be based on safety reviews of groups 1 and 2. There will be external safety monitoring reviews by the Data Safety Monitoring Committee (DSMC) between the two dose escalation groups and at a further two time points in group 3.

### Randomisation

Randomisation of participants will be carried out by unblinded study staff who are independent from the blinded team and do not perform any postvaccination procedures (such as ongoing eligibility or safety review). A web-based randomisation system will be used.

### Blinding

This study will be conducted observer- and participant-blind from the time of randomisation until participant unblinding which will occur once the last participant has completed their final visit. Observer and participant blinding is required to minimise the risk of bias on the reporting of adverse events following the administration of vaccine.

### Study visits

Vaccine or placebo will be administered at 0, 2 and 6 months. Participants will be directly observed for a minimum of 60 min following vaccination and then asked to complete an e-diary of their symptoms daily for 7 days following each vaccination. An in-person postvaccination review will occur 7 and 28 days following each vaccine when participants will be reviewed for any possible adverse events. There are a total of 12 scheduled study visits and participants will be followed up for 1 year following first vaccination.

### Recruitment and eligibility

Potential participants may be contacted by media advertisements, direct mail out or social media using an approved invitation letter or other approved advertising material to invite them to participate in the study. Participants will be reimbursed for their time, travel and inconvenience. Healthy adults between the ages 18–55 years inclusive will be eligible for enrolment. Individuals will be initially screened for eligibility by telephone followed by face-to-face visits at the trial centre. Screening visits will involve obtaining informed consent (see SALVO Informed Consent Form in online supplemental

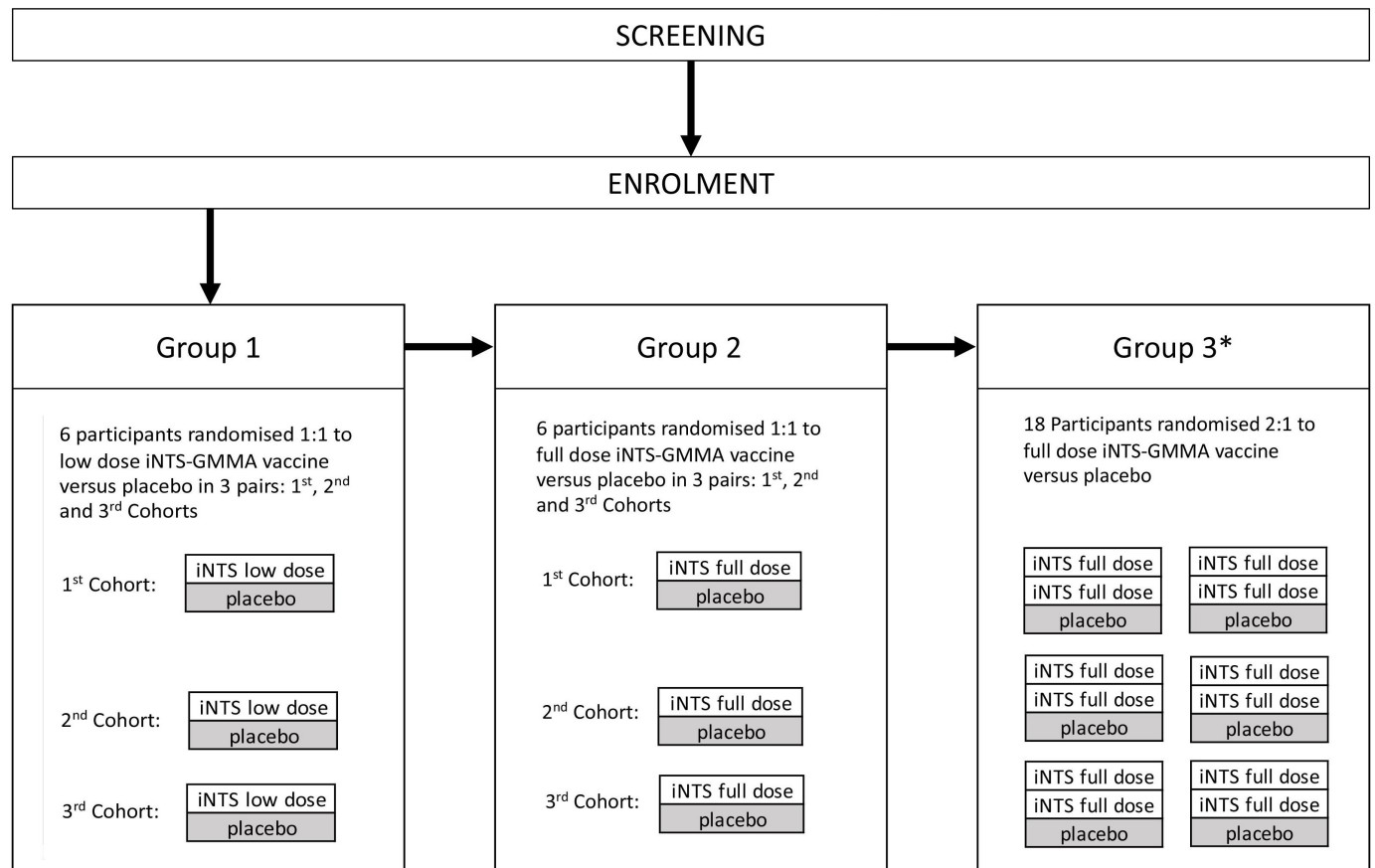

**Figure 2** *Salmonella* Vaccine Study in Oxford study design. *Decision to proceed to low or full dose will be based on safety reviews of groups 1 and 2. GMMA, Generalised Modules for Membrane Antigens; iNTS, invasive non-typhoidal *Salmonellosis*.

material 2), application of inclusion and exclusion criteria (summarised in table 2) and clinical eligibility assessments including vital signs, physical examination, baseline blood tests and urinalysis (SALVO Protocol in online supplemental material 1).

### Sample size and statistical analysis

The sample size in this study is 30–42 participants to account for additional participants to be recruited on DSMC advice. As the initial phase I trial primary objective is safety and tolerability, this sample size has been chosen to evaluate early data on adverse events associated with increasing dose level, with a larger subsequent phase I trial in a highly endemic country planned dependant on the trial safety data. The statistics for the primary endpoint are descriptive, with no testing of statistical significance. The CI will be set at 95%.

### Ethics and dissemination

As a first-in-human vaccine trial, the iNTS-GMMA vaccine has undergone appropriate preclinical toxicology studies indicating a well-tolerated vaccine. Participants will be actively monitored for their safety during the trial by review of an electronic diary, visits, clinical observations and safety blood tests and will have access to a 24-hour medical contact number. Appropriate risk and benefits of the study will be communicated to the participants, and

informed consent will be taken prior to any study related procedures. Local and national guidelines on confidentially and data protection will be adhered to.

The DSMC consisting of an experienced group of clinicians and a statistician will be appointed to provide real-time independent oversight of safety and trial conduct. The DSMC will review safety data collated from participant and clinician recorded entries including solicited and unsolicited adverse events, laboratory results and vital signs. Progression of enrolment from group 1 to group 2 to group 3 will only occur after DSMC review of the relevant safety data. Further DSMC reviews will occur regularly throughout the trial. A development safety update report for the IMP will be prepared annually, on the anniversary of the Medicines and Healthcare products Regulatory Agency (MHRA) approval for the trial.

Undertaking non-COVID-19 research during a dynamic COVID-19 pandemic represents a major logistical challenge. The safety of participants remains paramount and good infection prevention and control practices will be followed throughout the trial. The SALVO study team will monitor cases of COVID-19 within the participant cohort and the local population and will follow current national guidelines on COVID-19 with options including switching to phone appointments, halting or extending the trial.

**Table 2** Summary of *Salmonella* Vaccine Study in Oxford inclusion and exclusion criteria

| | |
|---|---|
| Inclusion criteria | Willing and able to give informed consent for participation in the study |
| | Aged between 18 and 55 years inclusive |
| | In good health as determined by:<br>► Medical history<br>► Physical examination<br>► Laboratory assessment<br>► Clinical judgement of the investigators |
| | Willing to use highly effective contraception from 1 month prior to receiving the first vaccine and for the duration of the study (females) |
| | Able to attend the scheduled visits and to comply with all study procedures, including internet access for the recording of diary cards |
| | Willing to allow his or her general practitioner and/or consultant, if appropriate, to be notified of participation in the study |
| | Willing to provide their national insurance number or passport number to be registered on The Over-Volunteering Prevention System (TOPS) |
| Exclusion criteria | History of significant organ/system disease that could interfere with the trial conduct or completion in the clinical judgement of the investigators |
| | Have any known or suspected impairment or alteration of immune function |
| | Study significant abnormalities on screening investigations that are either unlikely to resolve or do not resolve on repeat testing |
| | Prior history of receipt of an oral typhoid vaccine (eg, Ty21a) within the last 3 years or a paratyphoid vaccine (as part of a clinical trial) |
| | Prior history of participation in a typhoid or paratyphoid controlled human infection study* |
| | Receipt of a live vaccine within 4 weeks prior to vaccination or a killed vaccine within 7 days prior to vaccination |
| | Plan to receive any vaccine other than the study vaccine within 4 weeks after any study vaccination (except for COVID-19 vaccines) |
| | History of allergy or anaphylaxis to a previous vaccine or vaccine components |
| | Receipt of immunoglobulin or any blood product transfusion within 3 months of study start |
| | Participation in another research study involving an investigational product or that which may compromise the integrity of the study |
| | Inability, in the opinion of the investigator, to comply with all study requirements including likelihood of successful venepuncture during the trial |
| | Female participants who are pregnant, breastfeeding/lactating or planning pregnancy during the course of the study |
| | Weight less than 50 kg or a BMI<18.4 kg/m$^2$ or a BMI>40 kg/m$^2$ |
| | Any other significant disease or disorder which, in the opinion of the investigator, may:<br>► Put the participants at risk because of participation in the study<br>► Influence the result of the study<br>► Impair the participant's ability to participate in the study |

*Future studies in typhoid or paratyphoid endemic countries may consider including a history of *S.* Typhi or *S.* Paratyphi natural infection as an exclusion. This was not included in this study as the risk of prior natural infection was considered low.

Once the trial has been completed including analysis of data, results will be published in a peer-reviewed journal and presented at conferences. The results of this study will directly impact the appropriateness of subsequent trials with a larger sample size to begin in a sub-Saharan country of high iNTS endemicity.

This study has been approved by South Central—Oxford A Research Ethics Committee on 28th April 2022.

### Patient and public involvement

The protocol, study information booklet and recruitment materials were reviewed by a local patient consultation group who provided feedback and comments on the initial documents. Their comments led to changes in the participant-facing documents, ensuring they are easy and clear for participants to understand (please see SALVO Participant Information Sheet in online supplemental material 3).

## Study dates

Study recruitment began in May 2022. The estimated last participant last visit will be in December 2023.

## DISCUSSION

This will be the first phase I study investigating the iNTS-GMMA vaccine. The data generated by this trial will guide future vaccine development using this GMMA technology and may contribute to the licensure of the first vaccine against invasive non-typhoidal *Salmonella* species.

iNTS disproportionately causes severe disease in children under the age of 5 years in sub-Saharan Africa, and an effective vaccine will be of greatest benefit in this vulnerable population. As the UK is a country with a low burden of non-typhoid *Salmonella* disease,[22 23] healthy adult volunteers recruited to the SALVO study will be unlikely to have pre-existing immunity to *S.* Typhimurium or *S.* Enteritidis. However, assessment of the immunogenicity of the iNTS-GMMA vaccine in an immunologically naïve cohort in SALVO will inform the decision to progress to a second larger phase I trial in a sub-Saharan African adult population where iNTS is endemic. This second study may provide insight into the vaccine-induced immunogenicity following pre-existing immunity in adults. Future studies will recruit African infants, in whom any future licensed vaccine is likely to be deployed. However without a known immune correlate of protection, the efficacy of the vaccine in prevention of iNTS disease cannot be determined without much larger field studies in endemic settings.

As GMMAs originate from the bacterial outer membrane, they contain both the immunodominant OAg and multiple membrane proteins in their native conformation.[19] Alongside measurement of OAg binding antibody after vaccination, this study will use functional antibody assays (including serum bactericidal activity), to interrogate the outer membrane protein specific responses. This may reveal conserved proteins present across multiple *Salmonella* serovars capable of eliciting pan-protective immune responses. As a first-in-human phase I trial, the study design prioritises the safety of the participants. The dose escalation design allows investigation of the safety of the vaccine in a small cohort and at a significantly lower dose prior to escalation to the full dose vaccine. Multiple DSMC and internal safety reviews at set intervals provide further mandatory time points to formally review the safety data in addition to real-time monitoring by the study team. This is a participant-observer blind, randomised, placebo-controlled study. Blinding of the trial aims to reduce both recall and observational bias, which is intended to allow confidence in the study's final outcome. The use of a placebo-controlled study design allows a direct comparison between active vaccine and placebo groups aiming to account for potentially confounding factors, such as the incidence of SARS-CoV-2 infection during the study.

SALVO will be the first clinical trial of the iNTS-GMMA vaccine. Currently, the only other iNTS-based vaccines to enter clinical trials are two trivalent vaccines covering *S. enterica* serovars Typhimurium, Enteritidis and Typhi,[24 25] with results to be published. A live attenuated oral *S.* Typhimurium was trialled in participants in 2009 but has not progressed further in the intervening years.[26] Further iNTS-based vaccines are progressing through the preclinical phase including flagellin and OmpD-based vaccines.[27]

The study protocol was prepared in accordance with the SPIRIT 2013 Checklist.[28]

**Collaborators** Vacc-iNTS consortium collaborators: Francis Agyapong (Kwame Nkrumah University of Science and Technology Kumasi); Gianluca Breghi (Fondazione Achille Sclavo); John A. Crump (University of Otago); Fabio Fiorino (University of Siena); Melita A. Gordon (University of Liverpool); Jan Jacobs (Institute of Tropical Medicine Antwerp); Samuel Kariuki (Kenya Medical Research Institute); Stefano Malvolti (MM Global Health Consulting); Carsten Mantel (MM Global Health Consulting); Christian S. Marchello (University of Otago); Florian Marks (University of Cambridge and International Vaccine Institute); Donata Medaglini (Università di Siena and Sclavo Vaccines Association); Esther M. Muthumbi (KEMRI-Wellcome Trust Research Programme); Chisomo L. Msefula (University of Malawi); Tonney S. Nyirenda (University of Malawi); Robert Onsare (Kenya Medical Research Institute); Ellis Owusu-Dabo (Kwame Nkrumah University of Science and Technology Kumasi); Elena Pettini (University of Siena); J. Anthony G. Scott (KEMRI-Wellcome Trust Research Programme); Bassiahi Abdramane Soura (University of Ouagadougou); Tiziana Spadafina (Sclavo Vaccines Association); Bieke Tack (Institute of Tropical Medicine Antwerp).

**Contributors** BH and MR designed and authored the protocol. BH and NK wrote the manuscript. NO, CFDS, HR, RW, PF, UN, RC, AJP contributed to the protocol design and/or study set up.

**Funding** This research was funded in whole or in part by EU Framework Programme for Research and Innovation, Horizon2020, Vacc-iNTS no 815439 grant. For the purpose of Open Access, the author has applied a CC BY public copyright licence to any Author Accepted Manuscript (AAM) version arising from this submission.

**Competing interests** The iNTS-GMMA vaccine has been provided by the GSK Vaccines Institute for Global Health (GVGH). GVGH has reviewed the protocol developed by the Oxford Vaccine Group, University of Oxford and provided funding for clinical trial monitoring. PF, UN and RC were employees of the GSK Vaccines Institute for Global Health at the time in which the study was conducted. UN owns shares in GSK. GSK Vaccines Institute for Global Health Srl is an affiliate of GlaxoSmithKline Biologicals SA. This does not alter the authors' adherence to all journal policies on data and material sharing.

**Patient and public involvement** Patients and/or the public were involved in the design, or conduct, or reporting, or dissemination plans of this research. Refer to the Methods section for further details.

**Patient consent for publication** Not applicable.

**Provenance and peer review** Not commissioned; externally peer reviewed.

## ORCID iDs

Brama Hanumunthadu http://orcid.org/0000-0001-6263-2327
Andrew J Pollard http://orcid.org/0000-0001-7361-719X

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
