## [Reviewer comments · BMJ Open]

ARTICLE DETAILS

TITLE (PROVISIONAL)	Salmonella Vaccine Study in Oxford (SALVO) Trial: Protocol for an Observer-Participant Blind Randomised Placebo-Controlled Trial of the iNTS-GMMA Vaccine within a European Cohort
AUTHORS	Hanumunthadu, Brama; Kanji, Nasir; Owino, Nelly; Ferreira Da Silva, Carla; Robinson, Hannah; White, Rachel; Ferruzzi, Pietro; Nakakana, Usman; Canals, Rocio; Pollard, Andrew; Ramasamy, Maheshi; Vacc-iNTS Consortium, Collaborators

VERSION 1 – REVIEW

REVIEWER	Rasheed, Muhammad Asif COMSATS University Islamabad
REVIEW RETURNED	01-Apr-2023

GENERAL COMMENTS	Dear author, Please address the following comments. 1- How many persons will be there in each group? This is not mentioned.2- What about the gender of participating persons? Will there be equal numbers of males and females in each group?3- The plagiarism of the document is quite high.4- Why the first booster vaccine will be applied after 2 months? Should it not be applied after 1 month? Mostly, the titer of antibodies decrease after one month and a booster dose is required after 28 days or 1 month.5- I am confused about the statistical analysis. Why the statistical significance will not be tested?
---

REVIEWER	White, Aaron P University of Saskatchewan
REVIEW RETURNED	14-Apr-2023

GENERAL COMMENTS	Hanumunthadu et al. describe the design and implementation of a human clinical trial to test the tolerance of a vaccine against invasive Salmonellosis caused by S. enterica serovars Typhimurium and Enteritidis. At the basic form, patients will be tested with a low dose (5.3 ug of STm and 5.3 ug of SEN) and a high dose (20 ug of each). The vaccine form is the GMMA - generalized modules for membrane antigens - or membrane vesicles. I am familiar with the research leading up to this trial, in the publication of the animal studies examining GMMAs and their efficacy at blocking iNTS infections. The trial information is extensive but is well written and explained and the trial design is clearly laid out. I don't know all of the
--

	specifics that need to be reported for this type of publication but everything seems to be in order. The major comment I have is in Table 1 - the secondary outcome - "Measurement of serovar-specific O-antigen by ELISA before and after vaccination" I think the authors meant to say that they are measuring the levels of patient antibodies against STm and SEn O-antigen, which is a major immunogenic component of the GMMAs. This needs to be corrected and explained clearly in Table 1. Several Tables that are shown later have this explained correctly.
--	--

REVIEWER	Chaudhary, Shipra BP Koirala Institute of Health Sciences
REVIEW RETURNED	14-Jul-2023

GENERAL COMMENTS	It was a pleasure to review this important manuscript. I would be interested to know the results of this study. Here are a few comments/ suggestions from my end:  1. Exclusion criteria:  a. The authors have mentioned about oral typhoid vaccines and other typhoid studies in exclusion criteria (page 50 lines 32-34) but not specified clearly regarding those with previous typhoid infection or recipients of typhoid conjugate vaccine. Kindly clarify. b. Lines 43- What about the participants who require blood transfusion during the study period. This study involves a large volume of blood sampling (>600ml), which is of concern and certain participants might need transfusion in between. How will these be addressed? Please justify. 2. For COVID-19, if a participant gets COVID-19 vaccine in between the study period, not matching as per the study requirements, what will be done? Please specify. 3. Page 88 lines 7-8: Hb decrease from baseline unit is not clear- is it percentage or g/dl?
---

VERSION 1 – AUTHOR RESPONSE

Reviewer: 1

Please address the following comments.

1- How many persons will be there in each group? This is not mentioned.

Response: This is included in the study design section with participant numbers per group specified.

2- What about the gender of participating persons? Will there be equal numbers of males and females in each group?

Response: We are not controlling for this factor. The demographics will be published in any publication or report

3- The plagiarism of the document is quite high.

Response: Thank you for your comment. All references have been checked and are appropriate. The protocol will be included in the supplementary material. Please clarify if this answers your concern and if not – we would be grateful for which sections are of concern, with examples as we are unclear of the basis for this comment.

4- Why the first booster vaccine will be applied after 2 months? Should it not be applied after 1 month? Mostly, the titer of antibodies decrease after one month and a booster dose is required after 28 days or 1 month.

Response: This is a first in human study of the vaccine. The secondary objective is immunogenicity which is yet to be determined. The peak in antibody is typically considered to be at 28 days following vaccination. The 2-month boost has been chosen for two reasons. The first is at an appropriate interval to coincide with a presumed decline in antibody and to ensure longevity of response. The second is for pragmatic administering of the vaccine in the eventual target population and the local immunisation schedule.

5- I am confused about the statistical analysis. Why the statistical significance will not be tested?

Response: For the full dose vaccine and placebo, there will be 12 participants each. The numbers were considered too small to power appropriate statistical testing.

Reviewer: 2

Hanumunthadu et al. describe the design and implementation of a human clinical trial to test the tolerance of a vaccine against invasive Salmonellosis caused by *S. enterica* serovars Typhimurium and Enteritidis. At the basic form, patients will be tested with a low dose (5.3 ug of STm and 5.3 ug of SEn) and a high dose (20 ug of each). The vaccine form is the GMMA - generalized modules for membrane antigens - or membrane vesicles. I am familiar with the research leading up to this trial, in the publication of the animal studies examining GMMAs and their efficacy at blocking iNTS infections.

The trial information is extensive but is well written and explained and the trial design is clearly laid out. I don't know all of the specifics that need to be reported for this type of publication but everything seems to be in order.

The major comment I have is in Table 1 - the secondary outcome - "Measurement of serovar-specific O-antigen by ELISA before and after vaccination"

I think the authors meant to say that they are measuring the levels of patient antibodies against STm and SEn O-antigen, which is a major immunogenic component of the GMMAs. This needs to be corrected and explained clearly in Table 1. Several Tables that are shown later have this explained correctly.

Response: Thank you for your comments. We have updated the table to include this.

Reviewer: 3

It was a pleasure to review this important manuscript. I would be interested to know the results of this study. Here are a few comments/ suggestions from my end:

1. Exclusion criteria:

a. The authors have mentioned about oral typhoid vaccines and other typhoid studies in exclusion criteria (page 50 lines 32-34) but not specified clearly regarding those with previous typhoid infection or recipients of typhoid conjugate vaccine. Kindly clarify.

Response: In the study setting in the UK, participants were unlikely to have received a typhoid conjugate vaccine. In addition, as SEn and STm do not contain Vi polysaccharide (mostly) – it was felt that this vaccine would have minimal impact on the immunogenicity of the iNTS-GMMA vaccine immunogenicity outcome. Regarding prior Typhoid infection – in the study setting and local demographic, prior Typhoid disease outside of a number of Typhoid and Paratyphoid controlled human infection studies conducted in the last 10 years in the local area was considered unlikely.

b. Lines 43- What about the participants who require blood transfusion during the study period. This study involves a large volume of blood sampling (>600ml), which is of concern and certain participants might need transfusion in between. How will these be addressed? Please justify.

Response: This applies to any "planned" transfusion that is known prior to enrolment. If a participant required a blood transfusion after enrolment for clinical reasons that occurred during the study, then

this would take precedence for participant safety. However, significant anaemia requiring blood transfusion without any concomitant condition is extremely unlikely. A condition which predisposes to this would be an exclusion based on investigator judgement. To mitigate this concern, participants must meet a minimum threshold haemoglobin level prior to enrolment. As a comparison, blood sampling during the study is well below the maximum volume that can be donated during a single year under UK national guidance for the NHS blood donation service (i.e the volume of blood taken during the trial by itself would not be expected to be significant enough for the participants to require a blood transfusion).

2. For COVID-19, if a participant gets COVID-19 vaccine in between the study period, not matching as per the study requirements, what will be done? Please specify.

Response: The study protocol allows participants to receive the COVID-19 vaccine during the study and includes a specific section on this: '1.1.1 PARTICIPANTS INVITED FOR COVID-19 VACCINATION DURING THE TRIAL'. In short, we would plan the study vaccine to be given two weeks before or two weeks after the COVID vaccine.

3. Page 88 lines 7-8: Hb decrease from baseline unit is not clear- is it percentage or g/dl?

Response: This decline is based on g/dl and graded 1-4 based on the level of decrease as per the protocol included in the supplementary material.

VERSION 2 – REVIEW

REVIEWER	Chaudhary, Shipra BP Koirala Institute of Health Sciences
REVIEW RETURNED	02-Sep-2023

GENERAL COMMENTS	Thank you for your responses. I still believe it would be better to specify the additional exclusion criteria related to your responses to the comment 1 in the protocol too for clarity in global context.
--

VERSION 2 – AUTHOR RESPONSE

Comments to the Author:

Thank you for your responses.

I still believe it would be better to specify the additional exclusion criteria related to your responses to the comment 1 in the protocol too for clarity in global context.

Response to comment:

Thank you for your comment. The protocol has been approved by the REC and MHRA in the UK, and with participants enrolled on these exclusion criteria, therefore it is not possible to change the protocol at this stage. Future protocols for the investigation of vaccines in endemic countries (where it is most likely to have a greater bearing) may include this additional exclusion criterion, but this would be a consideration for the chief investigator of subsequent vaccination trials involving this vaccination. However, to better address your comment in the manuscript, I have included a footnote to the Criteria Table, which states: 'Future studies in Typhoid or Paratyphoid endemic countries may consider including a past history of S. Typhi or S. Paratyphi natural infection as an exclusion. This was not included in this study as the risk of prior natural infection was considered low.'